# Risk Factor Analysis of Ski and Snowboard Injuries During the 2023/2024 Winter Season: A Single, High-Volume Trauma Center Database Analysis

**DOI:** 10.3390/medicina61010117

**Published:** 2025-01-14

**Authors:** Michele Paolo Festini Capello, Pieralberto Valpiana, Giuseppe Aloisi, Giovanni Cristofolini, Svea Caren Misselwitz, Giuseppe Petralia, Mario Muselli, Salvatore Gioitta Iachino, Christian Schaller, Pier Francesco Indelli

**Affiliations:** 1Department Orthopaedic Surgery, Südtiroler Sanitätsbetrieb, Dantestraße 51, 39042 Brixen, Italysalvatore.gioittaiachino@sabes.it (S.G.I.); christian.schaller@sabes.it (C.S.);; 2Institute of Biomechanics, Paracelsus Medical University (PMU), 5020 Salzburg, Austria; 3Dipartimento di Medicina Clinica, Sanita’ Pubblica, Scienze della Vita e dell’Ambiente, Universita’ degli Studi dell’Aquila, 67100 L’Aquila, Italygiuseppe.petralia@graduate.univaq.it (G.P.);; 4The Breyer Center for Overseas Studies in Florence, Stanford University, 50125 Florence, Italy; 5Centro Eccellenza Sostituzioni Articolari Toscana, Department of Orthopaedic Surgery, Azienda Sanitaria Toscana Centro, 50054 Florence, Italy

**Keywords:** sports medicine, trauma, winter sports, ski and snowboard

## Abstract

*Background and Objectives:* The objective of the study was to evaluate the epidemiology of slope-related accidents in a high-volume trauma center during the winter season. In addition, this study aims to analyze patient-related, equipment-related, and environment-related characteristics. *Materials and Methods:* A questionnaire containing 22 items was distributed to all adult patients admitted to the emergency department of the Brixen Hospital (Italy) during the 2023/24 winter season because of a ski/snowboard-related injury. *Results:* The final database included 579 questionnaires from 579 patients who ultimately entered the study: 285 were females and 294 were males. The analysis of risk factors for fractures revealed that patients with fractures were generally older (50.7 ± 16.0 years) compared to those without fractures (45.4 ± 17.2 years, *p* = 0.0021). Thirty-six percent of injuries were considered as joint sprain/ligament strain: patients in this group were younger (45.5 ± 16.2 years) compared to all patients (48.3 ± 17.3 years, *p* = 0.0151). *Conclusions:* In conclusion, this study identified significant risk factors associated with skiing and snowboarding injuries. Fractures were more common among older and more experienced skiers, particularly those who described themselves as experts. Ligamentous distortions were more common among younger and less experienced skiers. Fatigue is generally underestimated, and the general physical preparation is often lacking for sports like skiing and snowboarding. Additionally, the absence of significant correlations between weather conditions, snow quality, equipment type, and the difficulty of the slope with injury risk suggests that individual factors such as age and skill level are more critical determinants of injury risk than environmental or equipment-related factors.

## 1. Introduction

Winter sports such as skiing and snowboarding are particularly popular in the Italian Dolomites, especially in South Tyrol. Data from the Provincial Institute of Statistics of Bozen, Italy (ASTAT) showed that during the 2022–2023 winter season [1], more than 11.000 ski/snowboard accidents occurred in the ski resorts in the province of Bozen, Italy. This number is in line with the previous 5 years, considering an exponential increase in skiers on the Dolomitian slopes [1,2,3,4,5] in the post-COVID-19 era. The risk of injury associated with skiing or snowboarding accidents is well documented in the scientific literature [6,7,8,9]. The most affected anatomical body parts are the lower extremities (particularly the knee), and the most reported injuries are anterior cruciate ligament (ACL) and medial collateral ligament (MCL) sprains, which have been reported between 10% and 33% of all skiing-related injuries [10,11,12]. While fatality rates have remained constant in recent decades [13], changes in equipment and protection gear have modified the historical injury types. While many studies analyzed typical risk factors such as age, gender, BMI, and skiing skills [14,15,16], little is known about other etiopathogenetic causes: environmental data (injury time, day of the week, altitude, slope difficulties, weather and slope conditions, ski model) and data regarding the characteristics of the traumatic event (cause, body district, exact diagnosis, required therapeutic plan, surgical indication) are still missing.

This study aimed to investigate the risk factors, category, and severity of skiing/snowboarding-related injuries and to identify possible patient- and environment-related risk factors: data acquired during the 2023/2024 ski season from the South Tyrol region of Italy were analyzed.

## 2. Materials and Methods

A questionnaire containing 22 items was distributed to all adult patients admitted to the emergency department of the SABES Brixen Hospital (Italy) during the 2023/24 winter season because of a ski/snowboard-related injury. All patients came from ski resorts across South Tyrol (Italy), including Alta Badia, Racines, Obereggen, Plan de Corones, Alta Pusteria, Val Gardena, Alpe di Siusi, Val Senales, Solda, and Val d’Ultimo.

The questionnaire (Figure 1), available in three languages (Italian, German, English), consisted of 22 items and investigated patient-related characteristics (sex, age, residence, height, weight, daily medications, comorbidities, level of expertise as a skier, frequency of skiing or snowboarding), environment-related characteristics (injury time, time spent skiing before the injury, day of the week, altitude, snow quality, slope grade, weather and slope conditions, ski model), and finally trauma-related characteristics (equipment type, cause, location of injury, type of transportation used to access the hospital). Inclusion criteria were age over 18 years and interest in participating in the study: all patients were admitted to the emergency room (ER) of the Brixen Hospital (Brixen, Italy) during the winter season 2023/2024 (8 December 2023–31 March 2024) for injuries that occurred on ski slopes. Exclusion criteria were age of less than 18 years and inability to fill the questionnaire due to linguistic problems. The questionnaire was distributed directly to the patients in the ER waiting area, except in emergencies where the questionnaire was given to the patients only after the initial injury assessment. While in the ER department, all patients underwent clinical evaluation by a specialist in emergency medicine followed by an orthopedic specialist examination. Standard radiological evaluation and CAT scans (if needed) were performed directly in the emergency room. Ligament lesions were all diagnosed clinically by senior orthopedic surgeons after careful clinical assessment and analysis of the mechanism of trauma: if needed, Magnetic Resonance Imaging (MRI directly) was requested and performed in the following days since it was not considered urgent. Clinical situations where the ligament lesion was not clinically detectable were not included in the ligament lesion count of this study. In addition, the attending doctor filled out the questionnaire with information related to the diagnosis, treatment, and prognosis. This questionnaire enabled the collection of comprehensive information about the trauma, allowing for the classification of the injury. All data were collected to describe the type of accident and to quantify the frequency and severity of these injuries in the entire study population.

### Statistical Analysis

All collected data were then transferred to a database for statistical analysis: qualitative variables were expressed as the number of cases or percentages, and the chi-squared test was used for comparisons between the groups; continuous variables were represented as mean and standard deviation (SD). All parameters were tested for normality with Kolmogorov–Smirnov test, and the rank-sum test or *t*-test were used to compare data between groups as appropriate. All injuries were classified into different groups based on the presence of a joint sprain, ligamentous strain, or ultimatel indication for surgery. Layered analysis by skill level was also conducted. Finally, multivariate linear regression analysis was performed with the stepwise method to identify independent variables correlated with joint sprain, ligamentous strain, or ultimately indication for surgery. For each analysis, an alpha level of 0.05 was considered statistically significant. The statistical analysis was performed using the STATA/BE 18 software for Windows.

## 3. Results

The final database included (Table 1) 579 questionnaires from 579 patients who ultimately entered the study: 285 were females and 294 were males. The mean age was 47.3 ± 16.9 years and the final cohort showed a mean body mass index (BMI) of 24.4 ± 3.6 kg/m^2^. The patients included in the study spent an average of 2.1 ± 0.8 days on ski slopes per year, and, on the injury date, were skiing/snowboarding for approximately 2.6 ± 1.7 h before the investigated accident happened. In the final cohort, when asked to self-assess their skills, almost two-thirds (355, 61.4%) of the patients described themselves as good skiers, one quarter (142, 24.5%) as experts, 79 (13.6%) as beginners, and 0.5% as professional athletes.

Interestingly (Table 2), most of the accidents happened on slopes of medium (red slopes) difficulty (49.9%), while 34.7% of patients hurt themselves on easy slopes (blue slopes) and only 11.2% on difficult slopes (black slopes); in addition, 1.4% of the patients reported that the trauma happened in “contained” snow parks, and 2.8% of patients reported an injury during “off-piste” or “off-trail” skiing or snowboarding. Surprisingly, in most cases (59.9%), the ski slopes were well prepared at the time of the injury (Table 2), while in 16.2% of the cases, the traumatic event happened on icy slopes, and in 17.6% of the cases, the slope was considered as “bumpy” at the time of the injury.

The weather conditions (Table 2) at the time of the injury were also mostly good (69.4% sunny and 22.1% cloudy), with only 5.9% of the accidents happening with fog or poor visibility and only 2.6% snowing.

Regarding the equipment (Table 2) used at the time of the injury, 60.3% of patients owned the equipment. The most used types of skies were all-around/easy-carver skies (51.6%), followed by race-carver skies (18.1%) and slalom-carver skies (11.7%); only 2.1% of the injuries occurred to snowboarders.

### 3.1. Injury Patterns and Causes

This section provides a detailed analysis of the types of injuries, their causes, and the most commonly affected anatomical sites observed in our study. The most frequent cause of accidents reported was a patient’s error (70%): collision with other skiers was reported in 13% of the cases, injuries during jumps in 7%, and fatigue-related injuries in 5.7% of the cases (Table 3). Other reported causes of injury included cause of matter (3.4%) and chair lift-associated accidents (1.2%). The most common clinical and radiological diagnosis was a ligament strain in 213 (36.8%) cases, while a fracture was diagnosed in 207 (35.7%) of the cases: a simple contusion occurred in 84 (14.5%) of the cases, a concussion in 40 (6.9%), and a joint dislocation in 23 (4.0%) of the cases. The most common injury sites were the following: the lower limb (55.6%), followed by the upper limb (24.7%), the trunk (11.4%), and finally the head (8.3%). After the initial assessment in the ER, 21.1% of the patients were directly admitted to the Orthopedic Department, while 38.17% ultimately required surgery.

### 3.2. Age and Risk Factors for Fractures (Table 4)

This section explores the relationship between age, skiing experience, and the risk of fractures, highlighting key patterns observed among different groups. The analysis of risk factors for fractures revealed that patients with fractures were generally older (50.7 ± 16.0 years) compared to those without fractures (45.4 ± 17.2 years, *p* = 0.0021) and had also more years of skiing experience (26.6 ± 15.8 vs. 23.1 ± 16.0, *p* = 0.0071): interestingly, a higher proportion of expert skiers experienced fractures compared to novice and good skiers (*p* = 0.0142). Most fractures occurred in patients skiing on medium difficulty (red) slopes (47.34%) and among those who described themselves as expert skiers. Fractures were also significantly associated with surgical intervention (56.1% of patients with fractures required surgery, *p* < 0.0001) and hospitalization (71.3% of patients with fractures required hospitalization, *p* < 0.0001). Logistic regression analysis identified age (OR: 1.02, 95% CI: 1.01–1.03, *p* < 0.0001) and being an expert skier (OR: 1.57, 95% CI: 1.07–2.33, *p* = 0.023) as significant independent risk factors for fractures.

**Table 4 medicina-61-00117-t004:** Fractures.

Fracture	No	Yes	*p*-Value
Age, mean ± SD	45.4 ± 17.2	50.7 ± 16.0	0.0021
BMI, mean ± SD	24.3 ± 3.6	24.6 ± 3.8	0.3901
Years experience, mean ± SD	23.1 ± 16.0	26.6 ± 15.8	0.0071
Driving skills, n (%)			
Novice	21 (26.6%)	58 (73.4%)	0.0672
Good	122 (34.4%)	233 (65.6%)	0.3812
Expert	63 (44.4%)	79 (55.6%)	0.0142
Professional athlete	1 (33.3%)	2 (66.7%)	0.7083
Slope difficulty, n (%)			
Blue	122 (61.0%)	79 (39.0%)	0.7513
Red	191 (66.1%)	98 (33.9%)	
Black	42 (64.6%)	23 (35.4%)	
Off piste	11 (68.8%)	5 (31.3%)	
Snowpark	6 (75.0%)	2 (25.0%)	
Snow quality, n (%)			
Good condition	223 (64.3%)	124 (35.7%)	0.3703
Ice	66 (70.2%)	28 (29.8%)	
Mogul slope	57 (55.9%)	45 (44.1%)	
Deep snow	10 (71.4%)	4 (28.6%)	
Snowpark	7 (70.0%)	3 (30.0%)	
Off track	9 (75.0%)	3 (25.0%)	
Weather conditions, n (%)			
Sunny	261 (64.8%)	141 (35.2%)	0.4772
Cloudy	79 (61.4%)	49 (38.6%)	
Fog/poor visibility	20 (60.6%)	14 (39.4%)	
Snow	12 (80.0%)	3 (20.0%)	
Type of equipment, n (%)			
Racecarver	62 (16.7%)	43 (20.8%)	0.7753
Allround/Easy-Carver	200 (53.8%)	99 (47.8%)	
SlalomCarver	41 (11.0%)	27 (13.0%)	
Tourenski	21 (5.6%)	10 (4.8%)	
Telemark	10 (2.7%)	4 (1.9%)	
Twin Tip	27 (7.3%)	18 (8.7%)	
Snowboard	7 (1.9%)	5 (2.4%)	
Injury site, n (%)			
Head	48 (100.0%)	0 (0.0%)	<0.00012
Upper limb	73 (51.0%)	70 (49.0%)	
Lower limb	223 (69.3%)	99 (30.7%)	
Trunk	28 (42.4%)	38 (57.6%)	
Comorbidity, n (%)			
No	325 (93.9%)	184 (90.2%)	0.1072
Yes	21 (6.1%)	20 (9.8%)	
Surgery, n (%)			
No	275 (76.8%)	83 (23.2%)	<0.00012
Yes	97 (43.9%)	124 (56.1%)	
Hospitalization, n (%)—No	337 (73.7%)	120 (26.3%)	
Hospitalization, n (%)—Yes	335 (28.7%)	87 (71.3%)	

n, number; SD, standard deviation.

### 3.3. Joint Sprains and Ligament Strains (Table 5)

Thirty-six percent of injuries were considered as joint sprain/ligament strain: patients in this group were younger (45.5 ± 16.2 years) compared to all patients (48.3 ± 17.3 years, *p* = 0.0151). Interestingly, novice and good skiers were more likely to experience joint sprains/ligament strains compared to expert skiers (*p* = 0.0472). Most joint sprains/ligament strains affected the lower limb (59.3%, *p* < 0.0001). Logistic regression analysis identified younger age (OR: 0.98, 95% CI: 0.97–0.99, *p* = 0.012) as significant independent risk factors for joint sprains/ligament strains, while lower skill level (novice vs. expert: OR: 0.46, 95% CI: 0.31–0.68, *p* < 0.0001) appears to be a protective factor.

**Table 5 medicina-61-00117-t005:** Joint sprains and ligament strains.

Ligaments Distortion	No	Yes	*p*-Value
Age, mean ± SD	48.3 ± 17.3	45.5 ± 16.2	0.0151
BMI, mean ± SD	24.4 ± 3.6	24.4 ± 3.7	0.6261
Driving skills, n (%) Novice	46 (58.2%)	33 (41.8%)	0.3232
Good	219 (61.7%)	136 (38.3%)	0.3392
Expert	99 (69.7%)	43 (30.3%)	0.0642
Professional athlete	2 (66.7%)	1 (33.3%)	0.6943
Slope difficulty, n (%) Blue	133 (66.2%)	68 (33.8%)	0.7733
Red	176 (60.9%)	113 (39.1%)	
Black	41 (63.1%)	24 (36.9%)	
Off track	10 (62.5%)	6 (37.5%)	
Snowpark	6 (75.0%)	2 (25.0%)	
Snow quality, n (%) Good conditions	225 (64.8%)	122 (35.2%)	0.5773
Ice	56 (59.6%)	38 (40.4%)	
Mogul slope	66 (64.7%)	36 (35.3%)	
Deep snow	6 (42.9%)	8 (57.1%)	
Snowpark	6 (60.0%)	4 (40.0%)	
Off piste	7 (58.3%)	5 (41.7%)	
Weather conditions, n (%)—Sunny	252 (62.7%)	150 (37.3%)	0.2002
Cloudy	84 (65.6%)	44 (34.4%)	
Fog/poor visibility	24 (70.6%)	10 (29.4%)	
Snow	6 (40.0%)	9 (60.0%)	
Type of equipment, n (%)—Racecarver	69 (65.7%)	36 (34.3%)	0.0383
Allround/Easy-Carver	172 (57.5%)	127 (42.5%)	
SlalomCarver	49 (72.1%)	19 (27.9%)	
Tourenski	17 (54.8%)	14 (45.2%)	
Telemark	10 (71.4%)	4 (28.6%)	
Twin TIP	36 (80.0%)	9 (20.0%)	
Snowboard	9 (75.0%)	3 (25.0%)	
Injury site, n (%)—Head	48 (100.0%)	0 (0.0%)	<0.00013
Upper limb	122 (85.3%)	21 (14.7%)	
Lower limb	131 (40.7%)	191 (59.3%)	
Trunk	65 (98.5%)	1 (1.5%)	
Comorbidity, n (%)—No	328 (64.4%)	181 (35.6%)	0.0372
Comorbidity, n (%)—Yes	33 (80.5%)	8 (19.5%)	
Surgery, n (%)—No	232 (64.8%)	126 (35.2%)	0.3122
Surgery, n (%)—Yes	134 (60.6%)	87 (39.4%)	
Hospitalization, n (%)—No	246 (53.8%)	211 (46.2%)	<0.00013
Hospitalization, n (%)—Yes	120 (98.4%)	2 (1.6%)	

n, number; SD, standard deviation.

### 3.4. Risk Factors for No Correlation

While several variables were found to influence the risk of fractures and ligament injuries, this section focuses on factors that did not show a significant correlation with injury outcomes. The statistical analysis demonstrated that several factors were not significant risk factors for either fractures or joint sprains/ligament strains: in particular, weather conditions at the time of the injury (*p* = 0.6062), snow quality (*p* = 0.9703), and the type of equipment used (*p* = 0.1262) were not associated with an increased risk of injury. Additionally, the slope difficulty at the time of injury was not correlated with the risk of injury (*p* = 0.7733 for slope difficulty); similarly, the ownership of the equipment was not correlated with the risk of injury (rented vs. owned, *p* = 0.055).

## 4. Discussion

This study reports on the risk factor of skiing and snowboarding injuries in the South Tyrol region of Italy during the 2023–2024 winter season. Most of the injuries involved the lower limbs (55.6%) while upper limb injuries accounted for 24.7%; interestingly, trunk injuries (including chest and spine) accounted for 11.4%. Our data analysis revealed that ligament and joint injuries were the most prevalent (36.8%), while fractures occurred in 35.7% of cases.

The comparison of our data with the previous literature showed that the percentage, location, and types of injuries have changed in recent years due to various factors, such as improvements in ski equipment. First, our data suggested a decline in injury rate [17], probably related to improvements in ski equipment, especially boots and bindings, which offered better protection to the lower extremities. The location of injuries has also changed: in fact, we registered a reduction in the incidence of lower limb injuries and an increase in upper limb injuries when compared to previous studies [17,18,19]. Unfortunately, our data suggested an increase in the rate of knee injuries [17,18,19], a trend due to the use of more technically advanced boots that provided better ankle but inferior torsional protection to the knee [6,17,19,20,21,22]. When comparing skiers and snowboarders, our data showed that skiers were more prone to injuries with respect to snowboarders (96.6% skiers vs. 3.4% snowboarders), although the determination of exact injury rates in these two sports remains challenging [23,24,25]. Historically, snowboarding injuries are increasing [25,26,27]; interestingly, snowboarders are more prone to sustaining an acute injury with respect to skiers [27]. The difference between skiers and snowboarders mainly stems from the different injury mechanisms and technical equipment used [28,29].

Our study confirms that the lower limb is still the most affected injury anatomical location (55.6% in our series), and a knee sprain represents the single, most occurred injury type: 30.56% of our case series (similar to the literature [8,9,29,30]). Specifically, in snowboarders, upper limb injuries are historically more frequent [17,28]. On the other side, looking at the lower limb, the most common lower-extremity injuries in snowboarders affected the lower trunk (lumbar spine, pelvis, and hip) [25]. This difference is largely due to variations in equipment, stance, and fall mechanisms [29]. Ankle injuries appeared also to be more common in snowboarders [25].

Upper limb injuries in our study accounted for 24.7%, which represents a lower rate with respect to other authors (33%) [9,30]. In the beginners’ group of our series, the site of the injury was significantly different, with a higher proportion of upper limb fractures (47%) while lower limb injuries were more common among those who sustained fractures (69.3%) compared to non-fractures (30.7%) (SS). Interestingly, shoulder injuries (10.01% in our series) represented ~11% of all injuries in other series [31,32]: historically, injuries to the shoulder, elbow, forearm, and wrist were more prevalent in snowboarders than skiers [33,34]. Looking at other injury sites, trunk injuries accounted for 11.4%; historically, those injuries are four times more frequent in snowboarding than in skiing [22]. Head injuries, in our series, accounted for 8.3% of all injuries. This rate is lower than the rate reported by the US Consumer Product Safety Commission [35], which reported that head injuries accounted for 14% of all ski and snowboard injuries. This difference in the head injury rate between our series and other reports was probably due to the increase in helmet use among skiers and snowboarders in our area. Snowboarders were historically at a higher risk than skiers for sustaining head trauma, particularly concussions [25,36], however, some studies show conflicting results [37,38].

Looking at the type of injury, ligament strains and joint sprains, in general, were the most prevalent injury type in our series: 36.8% of all injuries. Age, equipment type, injury site, and presence of comorbidities were found to be significant factors for ligament strains: in our series, patients with ligament strains were younger and had fewer comorbidities. Fractures were the second most common injury, representing 35.7% of cases; interestingly, they occurred more often to experienced skiers with an average of 26.6 years of ski experience with respect to inexperienced skiers (SS). This finding suggests that experienced skiers, in the current series, underscored the importance of an accurate self-assessment and awareness of their limitations to mitigate the risk of injury. On the other side, among beginners, the likelihood of fractures was significantly associated with the difficulty of the slope, with more difficult slopes (e.g., black slopes and snow parks) being linked to a higher risk of fracture.

In the current series, fractures were also the most common reason for surgical interventions and hospitalizations (60% of all fractures required Open Reduction Internal Fixation—ORIF). The nearly equal distribution, in this case series, between ligament injuries and fractures suggests that both soft tissue and bone structures are at significant risk during skiing activities. This is likely attributable to the dynamic and high-impact nature of skiing, which places significant stress, particularly on the knees and ankles.

The injury site was also significantly associated with the need for surgery, with lower limb injuries showing a higher need for surgery (46.3% in this series). No other factors (snow quality, weather conditions, driving skills, type of equipment) showed a significant association with the need for surgery based on the current logistic regression model.

### Limitations

This study has several limitations. First, the authors acknowledge that less severe traumatic events were managed directly by on-site medical providers in first aid stations; because of that, the reported high rate of fractures in comparison to the total number of traumatic events registered by the authors could be slightly overestimated. Second, the study is based on consecutive cases registered in a single medical facility in a single ski season; nevertheless, the current data may represent an accurate estimate of traumatic events that an alpine hospital could expect and handle during a winter season. Another limitation of the study is represented by the fact that the final diagnosis of any encounter was elaborated after the execution of basic diagnostic tests such as standard radiograms, ultrasound examination, and computer tomography (CT); more advanced imaging (like Magnetic Resonance Imaging or MRI) was not available in the emergency department or outpatient clinic. Because of this, soft tissue and ligamentous injuries were diagnosed clinically but not confirmed by MRI.

## 5. Conclusions

In conclusion, this study identified significant risk factors associated with skiing and snowboarding injuries. Fractures were more common among older individuals, particularly those who described themselves as experts, while ligamentous strains were more common among younger and less experienced skiers. Interestingly, fatigue was generally underestimated at the time of the injury. Since physical preparation is often lacking in sports like skiing and snowboarding, these findings highlight the importance of tailored injury prevention strategies based on the skier’s age and skill level. Additionally, the absence of significant correlations between weather conditions, snow quality, equipment type, and slope difficulty with injury risk suggested that individual factors such as age and skill level were more critical determinants of injury risk than environmental or equipment-related factors.

## Figures and Tables

**Figure 1 medicina-61-00117-f001:**
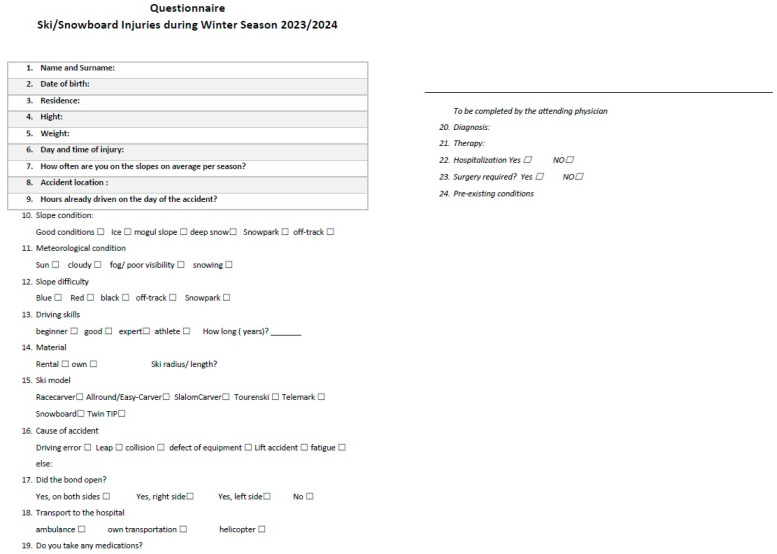
Questionnaire.

**Table 1 medicina-61-00117-t001:** Patient-related characteristics.

Total, n	579
Males, n (%)	294 (50.78%)
Females, n (%)	285 (49.22%)
Age, mean ± SD	47.3 ± 16.9
BMI, mean ± SD	24.4 ± 3.6
Hours of activity, mean ± SD	2.6 ± 1.7
Days of activity, mean ± SD	2.1 ± 0.8
Skills level, n (%)	
- Novice	79 (13.6%)
- Good	355 (61.4%)
- Expert	142 (24.5%)
- Professional athlete	3 (0.5%)

n, number; SD, standard deviation.

**Table 2 medicina-61-00117-t002:** Environment-related characteristics.

Slope Difficulty, n (%)	
- Blue	201 (34.7%)
- Red	289 (49.9%)
- Black	65 (11.2%)
- Off piste	16 (2.8%)
- Snowpark	8 (1.4%)
Snow quality, n (%)	
- Well prepared	347 (59.9%)
- Ice	94 (16.2%)
- Bucle	102 (17.6%)
- Fresh snow	14 (2.4%)
- Snowpark	10 (1.8%)
- “off track”	12 (2.1%)
Weather conditions, n (%)	
- Sunny	402 (69.4%)
- Cloudy	128 (22.1%)
- Fog/poor visibility	34 (5.9%)
- Snow	15 (2.6%)
Equipment, n (%)	
- Rented	230 (39,7%)
- Owned	349 (60,3%)
Type of equipment, n (%)	
- Racecarver	105 (18.1%)
- Allround/Easy-Carver	299 (51.6%)
- SlalomCarver	68 (11.7%)
- Tourenski	45 (7.8%)
- Twin Tip	45 (7.8%)
- Other	5 (0.9%)
- Snowboard	12 (2.1%)

n, number.

**Table 3 medicina-61-00117-t003:** Cause and type of injury.

Cause, n (%)	
- Mistake	400 (69.1%)
- Jump	42 (7.3%)
- Collision	77 (13.3%)
- Cause of matter	20 (3.4%)
- Chairlift	7 (1.2%)
- Fatigue	33 (5.7%)
Type of injury, n (%)	
- Fracture	207 (35.7%)
- Commotion	40 (6.9%)
- Contusion	84 (14.5%)
- Ligaments/joint	213 (36.8%)
- Luxation	23 (4.0%)
- Wound	9 (1.5%)
- PNX	2 (0.4%)
- Abdominal	1 (0.2%)
Injury site, n (%)	
- Head	48 (8.3%)
- Upper limb	143 (24.7%)
- Lower limb	322 (55.6%)
- Trunk	66 (11.4%)

n, number; PNX pneumothorax.

## Data Availability

The data will be available upon written request to the senior author.

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
