# Peer review of "Risk Factor Analysis of Ski and Snowboard Injuries During the 2023/2024 Winter Season: A Single, High-Volume Trauma Center Database Analysis"

_medicina, 2025, doi:10.3390/medicina61010117_

Round 1
Reviewer 1 Report
Comments and Suggestions for Authors
This study was very interesting, and I gained a lot of insights from reading it. I have a few comments for improvement that I hope will be helpful.
1. Please revise the title of Table 1, as “Multivariate analysis” does not accurately reflect its content.
2. The proportion of “Good” in “Driving skill” differs between the Results section (61.4%) and Table 1 (61.3%). Please ensure consistency.
3. In Table 2, the percentages under “Equipment” use commas (e.g., 39,7%). Please change them to dots for consistency.
4. In Table 3, “Cause of matter 20 (3.4%)” differs from the description in the text, “spontaneous opening of the binding (3.5%).” It might be clearer to unify these terms and percentages.
5. “Skiing experience” is mentioned in the text (Line 165) but is not included in Table 4. If it was omitted by mistake, please add it.
6. The percentages for “Novice” in Driving skills in Table 4 show a clear difference (No: 26.6%, Yes: 73.4%), but the p-value does not indicate statistical significance. Could you confirm if this data is correct?
7. In the text (Line 167-168), it states that “Most fractures occurred in patients skiing on medium difficulty (red) slopes (66.1%),” but in Table 4, the percentage for “Red slope” under “No” is 66.1%. Could you confirm if this is correct?
8. The text mentions “71.3% of patients with fractures required hospitalization,” (Line 170-171) but this data is not shown in Table 4. Including it there might improve consistency.
9. Tables 2-5 lack titles. Adding titles similar to Table 1 would improve clarity and consistency.
Reviewer 2 Report
Comments and Suggestions for Authors
This study is a single center cross-sectional study of risk of factors for ski and snowboarding injury in a single trauma center. The study design is clear, results are mostly presented clearly. Following are some of the comments to be addressed:
Title should be changed to “ Risk factor analysis of …..” because the paper mainly described the risk factors of different injuries not the incidence of these injuries among all the ski or snowboarding population. The populations analyzed are injured patients not all skiers or snowboarders. It cannot be incidence.
Line 62, “ascertain the incidence” may be changed to “investigate the risk factors, category, and severity…”
Lines 51, 78, “Body district” may be changed to “body parts” or “location of injury”
Table 1, 5: Why authors list “Driving skills” while they actually presented ski skills. Please change.
Each table should have a title and place in the top middle of the table.
Line 181-182, line break issue. Please correct.
Line 184-185, Authors stated Novice is risk factor, but the OR=0.45. Usually, in epidemiology, if a factor is risk factor, the OR should be >1, when a factor is a protective factor, OR <1. Why the authors statement and the OR values are opposite?
The results section has several subtitle that only have one word. Authors should separate them with a statement for each subsection, not just “Injuries, “age” and so on.
Line 192, Authors said the slope difficulty was not correlated with the risk injury, but P=0.021. Please check for accuracy.
Line 196, “Incidence should be changed to risk factor.
Lines 202, “Incidence” should be “percentage”.
Some of the percentage values have “,” not “.”. Please correct (lines 213, 220, 231).
